# Transthoracic Echocardiography-Based Prediction Model of Adverse Event Risk in Patients with COVID-19

Elena Zelikovna Golukhova [1], Inessa Viktorovna Slivneva [2,*], Maxim Leonidovich Mamalyga [3], Damir Ildarovich Marapov [4], Mikhail Nikolaevich Alekhin [5], Mikhail Mikhailovich Rybka [6] and Irina Vasilevna Volkovskaya [7]

1   A.N. Bakulev National Medical Scientific Center for Cardiovascular Surgery, Ministry of Health of the Russian Federation, 121552 Moscow, Russia; egolukhova@bakulev.ru
2   Department of Emergency Ultrasound and Functional Diagnostics, A.N. Bakulev National Medical Scientific Center for Cardiovascular Surgery, Ministry of Health of the Russian Federation, 121552 Moscow, Russia
3   Department of Surgical Treatment of Coronary Heart Disease, A.N. Bakulev National Medical Research Center for Cardiovascular Surgery, Ministry of Health of the Russian Federation, 121552 Moscow, Russia; mamalyga83@mail.ru
4   Department of Public Health, Economics and Health Care Management, Kazan State Medical Academy—Branch Campus of the Federal State Budgetary Educational Institution of Further Professional Education «Russian Medical Academy of Continuous Professional Education» of the Ministry of Healthcare of the Russian Federation, 420012 Kazan, Russia; damirov@list.ru
5   Functional Diagnostics Department of the Central Clinical Hospital with Polyclinic of the Russian Presidential Administration, 121359 Moscow, Russia; amn@mail.ru
6   Department of Anesthesiology and Intensive Care, A.N. Bakulev National Medical Scientific Center for Cardiovascular Surgery, Ministry of Health of the Russian Federation, 121552 Moscow, Russia; mmrybka@bakulev.ru
7   Polyclinic Department of the Institute of Coronary and Vascular Surgery, A.N. Bakulev National Medical Scientific Center for Cardiovascular Surgery, Ministry of Health of the Russian Federation, 121552 Moscow, Russia; ivvolkovskaya@bakulev.ru
*   Correspondence: slivneva@mail.ru

**Abstract:** Cardiopulmonary disorders cause a significant increase in the risk of adverse events in patients with COVID-19. Therefore, the development of new diagnostic and treatment methods for comorbid disorders in COVID-19 patients is one of the main public health challenges. The aim of the study was to analyze patient survival and to develop a predictive model of survival in adults with COVID-19 infection based on transthoracic echocardiography (TTE) parameters. We conducted a prospective, single-center, temporary hospital-based study of 110 patients with moderate to severe COVID-19. All patients underwent TTE evaluation. The predictors of mortality we identified in univariate and multivariable models and the predictive performance of the model were assessed using receiver operating characteristic (ROC) analysis and area under the curve (AUC). The predictive model included three factors: right ventricle (RV)/left ventricle (LV) area (odds ratio (OR) = 1.048 per 1/100 increase, *p* = 0.03), systolic pulmonary artery pressure (sPAP) (OR = 1.209 per 1 mm Hg increase, *p* < 0.001), and right ventricle free wall longitudinal strain (RV FW LS) (OR = 0.873 per 1% increase, *p* = 0.036). The AUC-ROC of the obtained model was 0.925 ± 0.031 (95% confidence interval (95% CI): 0.863–0.986). The sensitivity (Se) and specificity (Sp) measures of the models at the cut-off point of 0.129 were 93.8% and 81.9%, respectively. A binary logistic regression method resulted in the development of a prognostic model of mortality in patients with moderate and severe COVID-19 based on TTE data. It may also have additional implications for early risk stratification and clinical decision making in patients with COVID-19.

**Keywords:** COVID-19; TTE; transthoracic echocardiography; predictive model; multivariable model; right ventricle free wall longitudinal strain; systolic pulmonary artery pressure

## 1. Introduction

Nowadays, the rapid spread of the COVID-19 virus often results in comorbid pulmonary and cardiac dysfunction due to the high level of coronavirus tropism to respiratory system tissues and to vascular endothelium. Cardiopulmonary disorders significantly increase the risk of adverse events in COVID-19 patients. Therefore, one of the urgent public health problems is the development of new diagnostic and treatment methods for comorbid disorders in patients with COVID-19.

Coronavirus cytopathic action, inflammatory cytokine storms, and the cytokine effect on the myocardium, respiratory dysfunction and hypoxia, coagulation disorders, and disorders of the renin-angiotensin-aldosterone system are among the mechanisms underlying the development of cardiopulmonary anomalies in COVID-19 patients [1–3]. Increased pulmonary vascular resistance and the development of pulmonary hypertension in acute lung injury eventually lead to RV dysfunction [4]. It most often occurs as a consequence of acute respiratory distress syndrome (ARDS) [5,6] or in the context of acute pulmonary embolism following coagulation disorders and venous thromboembolism [1]. Even in the absence of cardiac abnormalities, COVID-19 progression leads to impaired central hemodynamic adaptation and exacerbates the patient's severe conditions [7,8]. Therefore, the development of new diagnostic strategies to detect pulmonary and cardiovascular dysfunction in patients with COVID-19 is an actual task.

The aim of the study was to investigate the survival rate and to develop a predictive model of survival in adults with COVID-19 infection based on TTE data.

## 2. Materials and Methods

### 2.1. Study Design and Population Profile

We conducted a prospective, single-center, temporary hospital-based study of patients with COVID-19 infection. From 148 patients in the initial data set with laboratory-confirmed COVID-19 infection, the final sample comprised 110 patients (74%). Patients with LV systolic dysfunction (i.e., an LV ejection fraction of less than 50%, signs of myocardial asynergy, or prior myocardial infarction), valve heart disease, prior heart surgery, percutaneous revascularization, a lung injury volume of less than 25%, evidence of RV or pulmonary artery outflow tract stenosis, non-adequate transthoracic acoustic window, or hemodynamic instability at time of study were excluded from the study. The local ethics committee approved the study project during the time of the temporary hospitalization (approval code number 3, dated 25 November 2021).

To reduce the risk of spreading infection, the study time was limited [9]. The treatment of patients differed depending on the period of the disease, clinical manifestation, leading pathogenetic syndrome, and concomitant diseases. In-hospital treatment tactics included non-pharmacological treatment (semi-bed rest, prone position of the body), oxygen therapy with possible respiratory support, and medication therapy, which included antiviral, anticoagulant, antiaggregant, anti-inflammatory, and antibacterial therapy.

We collected and analyzed transthoracic echocardiography (TTE) data. All patients also had chest computed tomography (CT), electrocardiography, and the required set of diagnostic and laboratory tests on admission. The median age of the patients was 63.0 years [interquartile range (IQR): 51.0; 74.0], and 57.3% were male.

The median time from the onset of the disease to hospitalization was 8 days, and the average duration of hospital stay was 13 days. Of the total patients, 11 (10.0%) were admitted to the intensive care unit (ICU) and 21 (19.1%) were transferred to the ICU due to the need for more active oxygen therapy. An analysis of the clinical profile of patients showed that most patients had a high risk of cardiovascular disease development: 81 patients (73.6%) had arterial hypertension, 20 patients (18.2%) had diabetes mellitus, and 18 patients (16.4%) had a history of cancer. Previous stroke or transient ischemic attack (present in 12.7% of the patients) and chronic obstructive pulmonary disease (COPD) (present in 10.9% of the patients) were also among the main comorbidities found. Laboratory tests showed

increased levels of biomarkers of systemic inflammatory response and of thrombosis. The clinical characteristics of the patients and TTE findings are presented in Table 1.

**Table 1.** Clinical characteristics and TTE characteristics of the overall cohort of patients with COVID-19.

| Variables | | Overall (*n* = 110) |
|---|---|---|
| Age, years | | 63 [51; 74] |
| Male | | 63 (57.3%) |
| BSA, m/m² | | 2.01 [1.85; 2.13] |
| Rhythm | Sinus | 89 (80.9%) |
| | Atrial fibrillation | 18 (16.4%) |
| | Pacemaker (without atrial fibrillation) | 3 (2.7%) |
| NEWS, scores | | 6 [5; 7] |
| SpO₂ on admission to the COVID-19 hospital, % | | 92 [91; 93] |
| Comorbidities | | |
| Hypertension | | 81 (73.6%) |
| Diabetes mellitus | | 20 (18.2%) |
| Cancer | | 18 (16.4%) |
| Encephalopathy at admission | | 15 (13.6%) |
| Stroke or transient ischemic attack | | 14 (12.7%) |
| COPD | | 12 (10.9%) |
| Bronchial asthma | | 9 (8.2%) |
| Chronic kidney disease | | 8 (7.3%) |
| Rheumatoid arthritis | | 3 (2.7%) |
| Current smoking | | 4 (3.6%) |
| TTE data | | |
| Maximum LA Vol (i), mL/m² | | 22.3 [19.1; 28.8] |
| LV EDI, mL/m² | | 49.6 [43.2; 58.1] |
| LV ESI, mL/m² | | 17.1 [14.3; 22.3] |
| LV SI, mL/m² | | 2.41 [2.00; 3.25] |
| LV EF, % | | 65 [60; 68] |
| CI, L/min/m² | | 2.44 [2.01; 3.20] |
| E/A | | 1.00 [0.80; 1.28] |
| E/e′ | | 7.72 [6.15; 9.80] |
| Maximum RA Vol (i), mL/m² | | 26.2 [19.9; 35.1] |
| Basal RV diameter, mm | | 40 [37; 43] |
| Mid cavitary RV diameter, mm | | 35 [30; 39] |
| RV longitudinal dimension, mm | | 60 [57; 66] |
| RV/LV area | | 0.64 [0.55; 0.74] |
| RV FAC, % | | 52.4 [45.1; 58.7] |
| TAPSE, mm | | 20 [18; 22] |
| Tricuspid annular S′ wave, cm/s (PW) | | 13 [11; 15] |
| RV FW LS, % (2D STE) | | 21.7 [16.2; 25.0] |
| sPAP, mm Hg | | 35 [30; 43] |

Abbreviations: BSA—body surface area; NEWS—National Early Warning Score; SpO₂—blood oxygen saturation; COPD—chronic obstructive pulmonary disease; LA—left atrium; Vol—volume; LV—left ventricle; EDI—end-diastolic volume index; ESI—end-systolic volume index; SI—stroke index; EF—ejection fraction; CI—cardiac index; RA—right atrium; RV—right ventricle; FAC—fractional area change; TAPSE—tricuspid annular plane systolic excursion; PW—pulse-wave; RV FW LS—right ventricle free wall longitudinal strain; sPAP—systolic pulmonary artery pressure. Data are expressed as number (percentage) or median [interquartile range] values.

The study sample was divided into two comparable groups depending on the outcome of the disease: survivors (*n* = 93) and non-survivors (*n* = 17).

### 2.2. Echocardiographic Analysis

Transthoracic echocardiographic examinations (TTE) were performed on a GE Vivid™ E9 ultrasound system (GE Vingmed Ultrasound AS, Horten, Norway) according to the approved protocol. Essential TTE positions (parasternal, apical, modified RV position, and subcostal) were used to visualize and evaluate the right heart. Dimensional and volumetric parameters of the left and right heart were measured in an apical four-chamber view with the calculation of the indexed parameters. LV volumes and ejection fractions were measured by biplane Simpson. Quantitative measurements were obtained according to the current recommendations of the American Society of Echocardiography and the European Association of Cardiovascular Imaging (ASE and EACVI, 2015) [10]. We recorded cine loops and images to reduce exposure time and enable subsequent remote analysis. Analysis of cine loops and images was conducted by two operators blinded to the clinical data.

All medical personnel were provided with protective equipment during the study in accordance with WHO standards [11] and the statement of protection [12]. To avoid possible virus transmission, ultrasonography was performed only on patients with confirmed COVID-19 infection. The ultrasound machine was cleaned as recommended after each patient [13,14].

LV diastolic function was assessed by measuring peak E velocity, calculation E/A, and additional parameters (E/e′, peak velocity of tricuspid regurgitation (TR), and maximum LA Vol $_{(i)}$) [15]. Transmitral flow was assessed by pulse-wave (PW) Doppler with measuring the peak E velocity and calculation E/A ratio. Additional parameters were required in the case of an E/A ratio of ≤0.8, along with a peak E velocity of >50 cm/s, or if the E/A ratio was within the range of 0.8 to 2.0. PW tissue Doppler imaging (TDI) was used to estimate the averaged early diastolic velocity of the mitral annulus movement (e′), which was determined at the position of the septal and lateral parts of the mitral valve.

Several parameters were assessed to analyze the RV contractile function. We used an apical modified RV view for tracing the RV diastolic and systolic areas. The change of the RV fractional area (RV FAC) was calculated according to the formula:

$$RV\ FAC\ (\%) = (RV\ EDA - RV\ ESA)/RV\ EDA \times 100\%,$$

where RV EDA is the end-diastolic area of RV and RV ESA is the end-systolic area of RV. The tricuspid annular plane systolic excursion (TAPSE) was determined in M-mode. The tricuspid annulus velocity (S′) was assessed by PW Doppler. To analyze the longitudinal deformation of the RV free wall—RV FW LS 2D STE (speckle-tracking echocardiography)—we used an apical modified RV view at a frame rate of >60 frames/s. The region of interest was selected with the subsequent correction of RV wall thickness. RV FW LS was expressed as an absolute value [10].

The flow of tricuspid regurgitation (TR) was assessed by color Doppler mapping, as well as by the jet density and contour characteristics in continuous-wave mode. The severity of the TR was ranged according to its significance: mild, moderate, or severe. The sPAP was determined by the peak velocity of the TR jet using Bernoulli's equation and adding the right atrial (RA) pressure value [16]. The mean pulmonary artery pressure (meanPAP) was estimated by using the maximal pulmonary regurgitation diastolic peak velocity [17] with added RA pressure. The RA pressure was assessed by measuring the maximum diameter and degree of collapse of the inferior vena cava.

### 2.3. Reproducibility

An intraclass correlation coefficient (ICC) was used to estimate the variability within a single operator (intra-observer variability), between different operators (inter-observer variability), and at different time points ("test-retest" 2 weeks after the initial analysis). Two observers independently estimated the pre-selected images of 15 random patients.

*2.4. Statistical Analysis*

All statistical analyses were performed using SPSS Statistics v.26 software (IBM Corporation). Continuous variables were presented as median and IQR values. Categorical variables were summarized using frequencies and percentages. The Mann–Whitney U test and chi-squared test were used to compare the two groups.

Simple logistic regression was used to assess the effect of each predictor on mortality. Next, a set of predictors based on the simple logistic regression Wald statistics was selected for further analysis in multiple logistic regression. The prognostic value of the multiple model was evaluated using ROC analysis.

Differences were considered statistically significant for *p*-values of <0.05.

## 3. Results

Hospital mortality among patients included in the study was 15.5% (*n* = 17). Patients in the non-survivors group were older (72 years [IQR: 60; 82] vs. 62 years [IQR: 50; 73]), had higher NEWS scores (7 [IQR: 6; 8] vs. 6 [IQR: 5; 7]), and a lower $SpO_2$ on admission (90% [IQR: 86; 92] vs. 93% [IQR: 92; 93]) (Table 2).

**Table 2.** Comparison of different parameters depending on survival status.

| Variables | Survivors (*n* = 93) | Non-Survivors (*n* = 17) | *p*-Value |
|---|---|---|---|
| Age, years | 62 [50; 73] | 72 [60; 82] | **0.046** |
| Male | 53 (57.0%) | 10 (58.8%) | 1.000 |
| BSA, $m/m^2$ | 1.99 [1.87; 2.10] | 2.03 [1.84; 2.18] | 0.738 |
| Sinus | 79 (84.9%) | 10 (58.8%) | **0.019** |
| Atrial fibrillation | 12 (12.9%) | 6 (35.3%) | **0.033** |
| Pacemaker | 2 (2.2%) | 1 (5.9%) | 0.399 |
| NEWS | 6 [5; 7] | 7 [6; 8] | **0.047** |
| $SpO_2$ on admission to a COVID-19 hospital, % | 93 [92; 93] | 90 [86; 92] | **0.002** |
| Comorbidities | | | |
| Hypertension | 66 (71.0%) | 15 (88.2%) | 0.230 |
| Diabetes mellitus | 16 (17.2%) | 4 (23.5%) | 0.508 |
| Cancer | 16 (17.2%) | 2 (11.8%) | 0.734 |
| Encephalopathy at admission | 10 (10.8%) | 5 (29.4%) | 0.055 |
| Stroke or transient ischemic attack | 8 (8.6%) | 6 (35.3%) | **0.008** |
| COPD | 9 (9.7%) | 3 (17.6%) | 0.393 |
| Bronchial asthma | 6 (6.5%) | 3 (17.6%) | 0.143 |
| Chronic kidney disease | 6 (6.5%) | 2 (11.8%) | 0.607 |
| Rheumatoid arthritis | 1 (1.1%) | 2 (11.8%) | 0.062 |
| Current smoking | 4 (4.3%) | 0 | 1.000 |
| Laboratory tests | | | |
| Monocytes, $\times 10^9/L$ | 0.49 [0.34; 0.63] | 0.45 [0.35; 0.63] | 0.908 |
| Neutrophils, $\times 10^9/L$ | 4.70 [3.16; 6.63] | 6.07 [3.42; 8.47] | 0.270 |
| Lymphocytes, $\times 10^9/L$ | 1.15 [0.92; 1.54] | 0.89 [0.60; 1.13] | **0.011** |
| D-dimer, ng/L | 572 [342; 859] | 1614 [385; 3187] | **0.048** |

**Table 2.** *Cont.*

| Variables | Survivors (*n* = 93) | Non-Survivors (*n* = 17) | *p*-Value |
|---|---|---|---|
| Hemoglobin, g/L | 140.3 [131.0; 148.1] | 130.9 [116.1; 142.1] | 0.088 |
| Erythrocytes, $\times 10^{12}/$L | 4.83 [4.48; 5.09] | 4.63 [4.36; 5.06] | 0.335 |
| White blood cells, $\times 10^{9}/$L | 6.6 [4.9; 9.2] | 9.5 [4.0; 11.8] | 0.169 |
| C-reactive protein, mg/L | 5.5 [2.4; 9.3] | 76.8 [35.3; 116.2] | **<0.001** |
| Platelets, $\times 10^{9}/$L | 213.5 [164.2; 260.4] | 148.8 [119.1; 198.1] | **0.004** |
| Lactate dehydrogenase, units/L | 288 [227; 387] | 422 [298; 665] | **<0.001** |
| Lung injury and oxygen therapy | | | |
| Lung tissue injury volume (CT data), % | 36 [28; 48] | 80 [64; 92] | **<0.001** |
| Nasal cannula (O$_2$ up to 15 L/min) | 81 (87.1%) | 0 | **<0.001** |
| High-flow nasal oxygen (AIRVO) | 5 (5.4%) | 0 | 1.000 |
| Non-invasive ventilation | 5 (5.4%) | 0 | 1.000 |
| Invasive mechanical ventilation | 2 (2.2%) | 17 (100%) | **<0.001** |
| VV ECMO | 0 | 4 (23.5%) | **<0.001** |
| Efferent therapy methods | | | |
| Plasmapheresis | 2 (2.2%) | 3 (17.6%) | **0.026** |
| Hemosorption | 2 (2.2%) | 3 (17.6%) | **0.026** |
| Complications | | | |
| Acute respiratory distress syndrome | 1 (1.1%) | 13 (76.5%) | **<0.001** |
| Systemic inflammatory response syndrome | 5 (5.4%) | 9 (52.9%) | **<0.001** |
| Acute heart failure | 2 (2.2%) | 11 (41.2%) | **<0.001** |
| Venous thrombosis | 9 (9.7%) | 2 (11.8%) | 0.678 |
| Multiple organ dysfunction | 1 (1.1%) | 9 (52.9%) | **<0.001** |
| Acute kidney injury | 1 (1.1%) | 4 (23.5%) | **0.002** |
| Disseminated intravascular coagulation | 0 | 4 (23.5%) | **<0.001** |
| Cerebral edema | 0 | 3 (17.6%) | **0.003** |
| Gastrointestinal bleeding | 0 | 1 (5.9%) | 0.155 |
| Haemorrhagic stroke | 1 (1.1%) | 0 | 1.000 |
| Characteristics of admission and hospital stay | | | |
| Days from illness onset to hospital admission, days | 8 [6; 11] | 9 [6; 10] | 0.533 |
| ICU admission | 5 (5.4%) | 6 (35.3%) | **0.002** |
| In-hospital transfer to the ICU | 10 (10.8%) | 11 (64.7%) | **<0.001** |
| In-hospital stay, days | 13 [12; 16] | 12 [8; 17] | 0.320 |
| Duration of ICU stay (only ICU patients), days | 1 [0; 3] | 9 [7; 13] | **<0.001** |

Abbreviations: BSA—body surface area; NEWS—National Early Warning Score; SpO$_2$—blood oxygen saturation; COPD—chronic obstructive pulmonary disease; ICU—intensive care unit; CT—computed tomography; AIRVO—humidifier with integrated air flow generator; VV—veno-venous; ECMO—extracorporeal membrane oxygenation. Data are expressed as number (percentage) or median [interquartile range] values. Bold indicates significance (*p*-value of <0.05).

Stroke or transient ischemic attack were the most common anamnestic factors among non-survivors and were reported in 35.3% of patients vs. 8.6% in survivors (*p* = 0.008). Non-survivors had significantly higher rates of life-threatening conditions, such as developed

ARDS (76.5% vs. 1.1%, $p < 0.001$), systemic inflammatory response (52.9% vs. 5.4%, $p < 0.001$), and acute heart failure (41.2% vs. 2.2%, $p < 0.001$).

A statistically significant difference in the level of the following laboratory tests was found in non-survivors: higher level of D-dimer, $p = 0.048$; lower platelets count, $p = 0.004$; high level of C-reactive protein, $p < 0.001$; higher level of lactate dehydrogenase, $p < 0.001$; and lower lymphocytes count, $p = 0.004$.

Chest CT scans confirmed a higher volume of lung injury in non-survivor patients (80% [IQR: 64; 92] vs. 36% [IQR: 28; 48]). Invasive mechanical ventilation was used in all patients in the non-survivors group. Transfer to extracorporeal membrane oxygenation (ECMO) for severe respiratory failure only occurred in the non-survivors group. Efferent methods of detoxification were used more frequently in the group of non-survivors (17.6% vs. 2.2%, $p = 0.026$).

The median time from onset to admission did not differ between the groups. Depending on baseline severity and respiratory support, non-survivors patients were more frequently admitted to the ICU (5.4% vs. 35%), were more frequently transferred to the ICU (10.8% vs. 64%), and had increased ICU lengths of stay ($p < 0.001$).

### 3.1. Echocardiographic Analysis

TTE parameters estimation in comparative analysis in the non-survivors group presented an increase in the dimensional and volumetric parameters of the right atrium (RA), RV dilatation at basal level (42 mm vs. 39 mm, $p = 0.004$), deterioration of RV contractile function (RV FAC: 49.0% vs. 53.4%, $p = 0.007$; TAPSE: 16 mm vs. 20 mm, $p < 0.001$; S wave: 12 cm/s vs. 13 cm/s, $p = 0.016$; RV FW LS: 15.2% vs. 22.3%, $p = 0.006$), increased pulmonary pressure (sPAP: 47 mm Hg vs. 35 mm Hg, $p < 0.001$; meanPAP: 23 mm Hg vs. 15 mm Hg, $p < 0.001$), and the prevalence of moderate or severe TR (Table 3). meanPAP was not determined in 39% of patients due to imaging limitations.

**Table 3.** Echocardiography of patients with COVID-19 infection.

| Variables | Survivors (n = 93) | Non-Survivors (n = 17) | p-Value |
|---|---|---|---|
| Minimum LA diameter, mm | 39 [36; 42] | 40 [38; 44] | 0.388 |
| Maximum LA diameter, mm | 52 [48; 56] | 60 [52; 61] | **0.014** |
| Maximum LA Vol $_{(i)}$, ml/m$^2$ | 21.6 [19.1; 28.7] | 25.6 [21.9; 31.0] | **0.088** |
| LV EDI, mL/m$^2$ | 49.3 [43.2; 56.6] | 51.4 [43.7; 68.3] | 0.370 |
| LV ESI, mL/m$^2$ | 16.7 [14.2; 22.1] | 18.0 [15.2; 28.7] | 0.239 |
| LV SI, mL/m$^2$ | 31.8 [27.2; 36.8] | 38.8 [30.2; 41.6] | 0.404 |
| LV EF, % | 65.0 [60.0; 69.0] | 63.0 [58.0; 65.0] | 0.116 |
| CI, l/min/m$^2$ | 2.39 [2.00; 3.09] | 3.02 [2.19; 3.71] | 0.171 |
| E/A | 0.94 [0.79; 1.20] | 1.40 [1.02; 1.87] | **0.009** |
| E/e′ | 7.56 [6.00; 9.14] | 9.87 [8.40; 12.29] | **0.001** |
| LV septal wall thickness, mm | 13 [12; 16] | 15 [13; 17] | 0.215 |
| Characteristics of the right heart chambers | | | |
| Minimum RA diameter, mm | 42 [37; 45] | 44 [43; 49] | **0.007** |
| Maximum RA diameter, mm | 51 [47; 55] | 56 [51; 61] | **0.004** |
| Maximum RA Vol $_{(i)}$, mL/m$^2$ | 24.7 [19.6; 34.8] | 32.2 [26.4; 42.5] | **0.002** |
| Basal RV diameter, mm | 39 [37; 43] | 42 [40; 48] | **0.004** |
| Mid cavitary RV diameter, mm | 34 [30; 39] | 36 [33; 41] | 0.156 |
| RV longitudinal dimension, mm | 60 [57; 66] | 64 [61; 69] | 0.119 |

**Table 3.** *Cont.*

| Variables | Survivors (*n* = 93) | Non-Survivors (*n* = 17) | *p*-Value |
|---|---|---|---|
| RV/LV area | 0.64 [0.55; 0.73] | 0.70 [0.54; 1.05] | 0.228 |
| PA trunk diameter, mm | 25 [23; 27] | 25 [25; 30] | 0.396 |
| RV FAC, % | 53.4 [46.4; 60.2] | 49.0 [42.5; 53.1] | **0.007** |
| TAPSE, mm | 20 [19; 22] | 16 [16; 19] | **<0.001** |
| Tricuspid annular S′ wave, cm/s (PW) | 13 [12; 15] | 12 [9; 13] | **0.016** |
| RV FW LS, % (2D STE) | 22.3 [17.7; 26.2] | 15.2 [11.7; 18.3] | **0.006** |
| RV wall thickness (subcostal), mm | 5 [4; 6] | 6 [5; 6] | **0.009** |
| Pulmonary hemodynamic parameters | | | |
| sPAP, mm Hg | 35 [28; 39] | 47 [42; 55] | **<0.001** |
| meanPAP, mm Hg | 15 [11; 21] | 23 [20; 31] | **<0.001** |
| Inferior vena cava diameter, mm | 22 [19; 24] | 23 [20; 24] | 0.319 |
| TR moderate to severe | 14 (15.4%) | 7 (43.7%) | **0.015** |

Abbreviations: LA—left atrium; Vol—volume; LV—left ventricle; EDI—end-diastolic volume index; ESI—end-systolic volume index; SI—stroke index; EF—ejection fraction; CI—cardiac index; RA—right atrium; RV—right ventricle; PA—pulmonary artery; FAC—fractional area change; TAPSE—tricuspid annular plane systolic excursion; PW—pulse-wave; RV FW LS—right ventricle free wall longitudinal strain; sPAP—systolic pulmonary artery pressure; meanPAP—mean pulmonary artery pressure; TR—tricuspid regurgitation. Data are expressed as number (percentage) or median [interquartile range] values. Bold indicates significance (*p*-value of < 0.05).

### 3.2. Logistic Regression Analysis and Prognostic Model Quality Assessment

The impact of various predictors of cardiac status according to TTE data on mortality risk in patients with COVID-19 was assessed by binary logistic regression. The results of the univariate analysis are presented in Table 4.

**Table 4.** Results of the predictor impact assessment on mortality in patients with COVID-19 infection.

| Prognostic Factor | B | OR (95% CI) | *p*-Value |
|---|---|---|---|
| Minimum diameter LA, mm | 0.072 | 1.074 (0.969–1.191) | 0.173 |
| Maximum diameter LA, mm | 0.093 | 1.097 (1.019–1.182) | **0.014** |
| Maximum LA Vol $_{(i)}$, mL/m$^2$ | 0.068 | 1.070 (1.008–1.136) | **0.026** |
| LV EDI, mL/m$^2$ | 0.022 | 1.022 (0.983–1.064) | 0.272 |
| LV ESI, mL/m$^2$ | 0.059 | 1.06 (0.980–1.147) | 0.145 |
| LV SI, mL/m$^2$ | 0.018 | 1.018 (0.955–1.085) | 0.585 |
| LV EF, % | −0.060 | 0.942 (0.870–1.021) | 0.144 |
| CI, L/min/m$^2$ | 0.416 | 1.516 (0.775–2.964) | 0.224 |
| E/A | 1.099 | 3.001 (1.273–7.076) | **0.012** |
| E/e′ | 0.222 | 1.249 (1.079–1.445) | **0.003** |
| LV septal wall thickness, mm | 0.038 | 1.039 (0.872–1.237) | 0.670 |
| Minimum RA diameter, mm | 0.098 | 1.103 (1.016–1.198) | **0.019** |
| Maximum RA diameter, mm | 0.115 | 1.122 (1.039–1.211) | **0.003** |
| Maximum RA Vol $_{(i)}$, mL/m$^2$ | 0.074 | 1.076 (1.029–1.125) | **0.001** |

**Table 4.** *Cont.*

| Prognostic Factor | B | OR (95% CI) | *p*-Value |
|---|---|---|---|
| Basal RV diameter, mm | 0.129 | 1.138 (1.033–1.253) | **0.009** |
| Mid cavitary RV diameter, mm | 0.069 | 1.072 (0.993–1.157) | **0.075** |
| RV longitudinal dimension, mm | 0.046 | 1.048 (0.978–1.122) | 0.184 |
| RV/LV area, 1/100 | 0.041 | 1.041 (1.010–1.074) | **0.010** |
| PA trunk diameter, mm | 0.090 | 1.095 (0.937–1.279) | 0.256 |
| RV FAC, % | −0.069 | 0.933 (0.885–0.983) | **0.010** |
| TAPSE, mm | −0.558 | 0.572 (0.429–0.764) | **<0.001** |
| Tricuspid annular S′ wave, cm/s (PW) | −0.230 | 0.794 (0.656–0.962) | **0.019** |
| RV FW LS, % (2D STE) | −0.122 | 0.885 (0.808–0.970) | **0.009** |
| RV wall thickness (subcostal), mm | 0.718 | 2.051 (1.123–3.745) | **0.019** |
| sPAP, mm Hg | 0.169 | 1.184 (1.093–1.282) | **<0.001** |
| meanPAP, mm Hg | 0.168 | 1.183 (1.082–1.293) | **<0.001** |
| Inferior vena cava diameter, mm | 0.099 | 1.104 (0.906–1.346) | 0.325 |
| TR, moderate to severe/mild | 1.575 | 4.831 (1.535–15.207) | **0.007** |

Abbreviations: B—regression coefficient; OR—odds ratio; 95% CI—95% confidence interval; LA—left atrium; Vol—volume; LV—left ventricle; EDI—end-diastolic volume index; ESI—end-systolic volume index; SI—stroke index; EF—ejection fraction; CI—cardiac index; RA—right atrium; RV—right ventricle; PA—pulmonary artery; FAC—fractional area change; TAPSE—tricuspid annular plane systolic excursion; PW—pulse-wave; RV FW LS—right ventricle free wall longitudinal strain; sPAP—systolic pulmonary artery pressure; meanPAP—mean pulmonary artery pressure; TR—tricuspid regurgitation. Bold indicates significance (*p*-value of < 0.05).

Further, the predictors were combined in a multivariable model to predict the mortality risk in a patient with COVID-19 based on TTE parameters of the right heart. Using binary logistic regression with factor selection by exclusion, significant factors were identified, and the following model was obtained, which showed statistical significance (*p* < 0.001) (Figures 1 and 2).

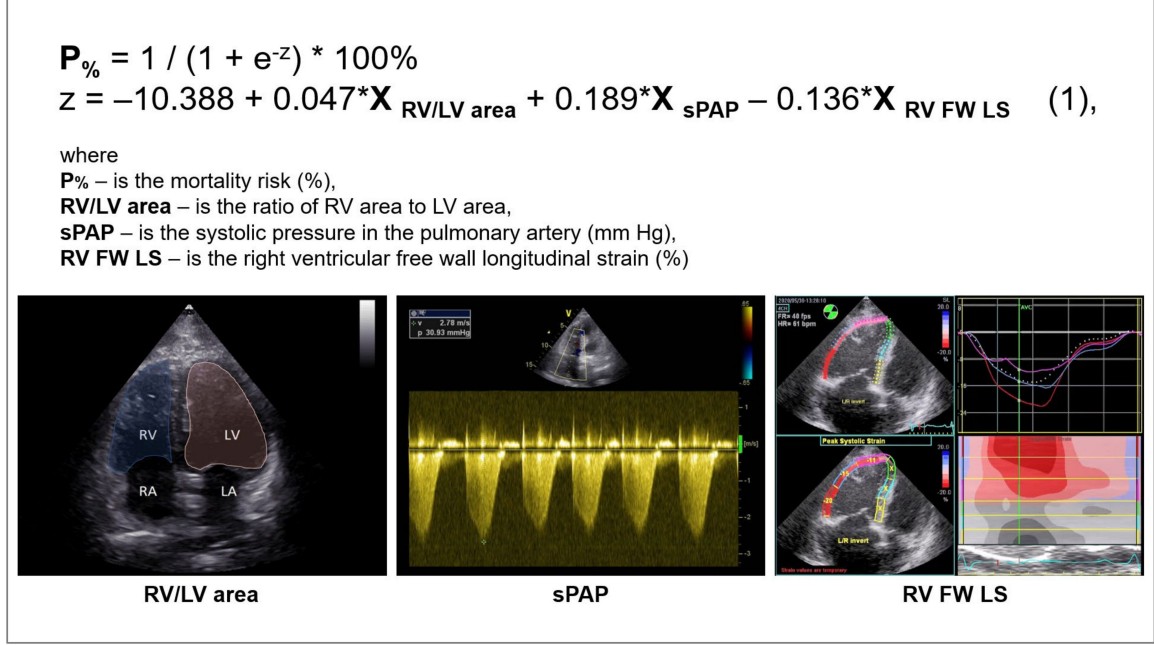

$$P_\% = 1 / (1 + e^{-z}) * 100\%$$
$$z = -10.388 + 0.047 * X_{RV/LV\ area} + 0.189 * X_{sPAP} - 0.136 * X_{RV\ FW\ LS} \quad (1),$$

where
**P%** – is the mortality risk (%),
**RV/LV area** – is the ratio of RV area to LV area,
**sPAP** – is the systolic pressure in the pulmonary artery (mm Hg),
**RV FW LS** – is the right ventricular free wall longitudinal strain (%)

RV/LV area                              sPAP                              RV FW LS

**Figure 1.** Multivariable model for predicting the risk of adverse outcomes in patients with COVID-19 infection.

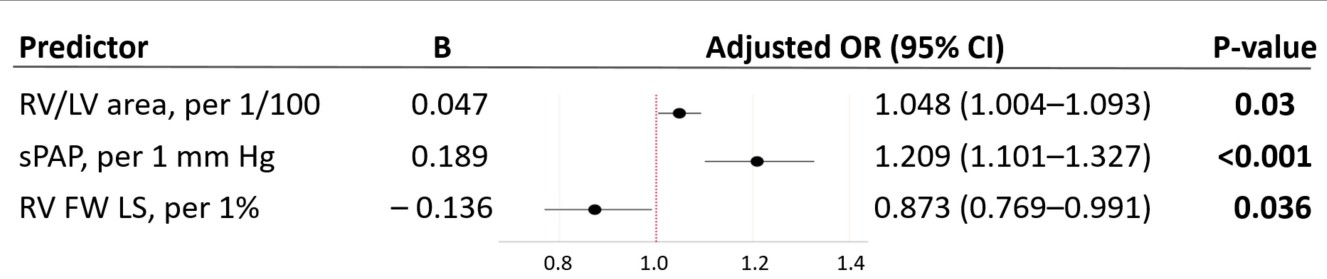

| Predictor | B | Adjusted OR (95% CI) | | P-value |
|---|---|---|---|---|
| RV/LV area, per 1/100 | 0.047 | | 1.048 (1.004–1.093) | **0.03** |
| sPAP, per 1 mm Hg | 0.189 | | 1.209 (1.101–1.327) | **<0.001** |
| RV FW LS, per 1% | − 0.136 | | 0.873 (0.769–0.991) | **0.036** |

**Figure 2.** Characteristics of predictors included in the model (1). B—value of the coefficient in the equation; OR—odds ratio of mortality; 95% CI—95% confidence interval; RV/LV area—right ventricle to left ventricle area ratio; sPAP—systolic pulmonary artery pressure; RV FW LS (2D STE)—right ventricle free wall longitudinal strain (two-dimensional speckle-tracking echocardiography). Bold indicates significance (*p*-value of < 0.05).

The variables RV/LV area and sPAP had a positive correlation with the mortality risk with OR = 1.048 per 1/100 increase in RV/LV area and OR = 1.209 per 1 mm Hg increase in sPAP. The longitudinal strain of the RV free wall had a negative correlation with the mortality risk (protective effect) with OR = 0.873 per 1% increase in RV FW LS. The threshold value of the logistic function *p*% was determined using the method of ROC-curve analysis. The resulting curve is shown in Figure 3.

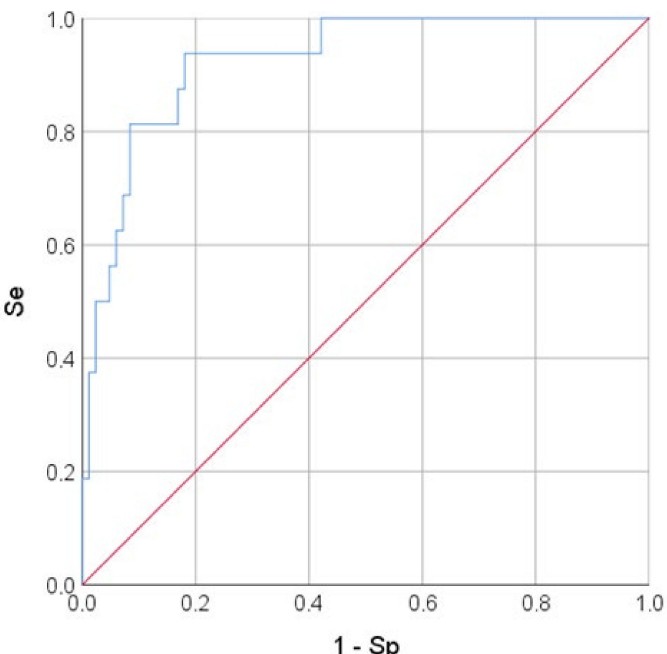

**Figure 3.** ROC-curve. The mortality risk on the values of the logistic function P%. The AUC-ROC was 0.925 ± 0.031 (95% CI: 0.863–0.986). The Se and Sp measures of model (1) at the cut-off point of 0.129 were 93.8% and 81.9%, respectively. The red line is the zero predictive value line (at Se + Sp = 100%), blue line–ROC-curve representing to the sensitivity function of the proportion of misclassified negative outcomes.

### 3.3. Assessment of Reproducibility

The variability of the test-retest, which includes intra-server and inter-server comparisons, is presented in Table 5.

**Table 5.** Results of intra-observer, inter-observer, and test-retest reproducibility analyses.

| Variable | ICC (95% Confidence Interval) | | |
|---|---|---|---|
| | **Intra-Observer** | **Inter-Observer** | **Test-Retest** |
| RV FW LS | 0.88 (0.68–0.96) | 0.84 (0.60–0.94) | 0.85 (0.63–0.95) |
| TR velocity | 0.96 (0.90–0.99) | 0.96 (0.89–0.98) | 0.95 (0.87–0.98) |
| RV area | 0.97 (0.92–0.99) | 0.93 (0.80–0.97) | 0.95 (0.86–0.98) |
| LV area | 0.98 (0.96–0.99) | 0.97 (0.92–0.99) | 0.98 (0.94–0.99) |

Abbreviations: ICC—intraclass correlation coefficient; RV FW LS—right ventricular free wall longitudinal strain; TR—tricuspid regurgitation; RV—right ventricle; LV—left ventricle.

## 4. Discussion

COVID-19 is not only a respiratory disease, but also a multisystem disease, and the combination of pulmonary and extrapulmonary symptoms is typically found in patients with severe COVID-19 [18]. A number of studies [19,20] have shown that COVID-19 significantly increases the risk of mortality because it causes a complex of interrelated disorders, worsening the course of cardiovascular diseases or provoking their occurrence.

The high level of comorbidity in the current study may reflect the age group of the patients. The median age of surviving patients was 62 years [IQR: 50; 73], and the age category of non-survivors was over 72 years [IQR: 60; 82] ($p = 0.046$). Older age has been reported previously as a risk factor for increased mortality in COVID-19 patients [21–23]. Patients in this age group had severe pneumonia, especially those with high blood pressure, coronary artery disease, and/or diabetes [24]. ARDS and other pulmonary complications, as well as multiple organ damage, are among the health problems reported in such patients [25].

In our study, hypertension was the most frequent anamnestic factor in both groups (71.0% in the survivors and 88.2% in the non-survivors). In addition to hypertension, diabetes mellitus, cancer, stroke or transient ischemic attack, and COPD dominated in the structure of comorbid pathology. However, statistical differences have been found only for a history of neurological disorders (stroke or transient ischemic attack). Prehospital neurological disorders were previously shown as predictors of high risk of adverse outcome [26]. The exacerbation of neurological symptoms in patients with COVID-19 may be due to the direct cytotoxic effect of the virus on the central nervous system, as well as mediated through thromboembolic and hypoxic damage, which causes cerebral edema [18].

Moreover, it has been shown in studies that cardiac rhythm disturbances occurring in COVID-19 may be due to presence of electrolyte and systemic hemodynamic disorders [27]. We found an increased incidence of atrial fibrillation in non-survivor patients with COVID-19 ($p = 0.033$). This is consistent with the results of the study conducted by Wang et al. [24], where 44% of patients with severe COVID-19 had arrhythmia.

According to Zaim et al. [19], an unfavorable COVID-19 prognosis mainly depends on the type of multiple organ failure. Analysis of the structure of multiple organ failure in our study showed that the main complications worsening the condition of patients with COVID-19 were acute respiratory distress syndrome, acute heart failure, and renal failure. These disorders exacerbated the severity of the systemic inflammatory response (increased C-reactive protein, $p < 0.001$) and hemostasis disorders (increased D-dimer, $p = 0.048$), which were also associated with an unfavorable prognosis.

Lung injury in COVID-19 patients can contribute to the imbalance of the ventilation–perfusion ratio, which, in turn, can cause a reduction in functional residual gas volume, leading to increased pulmonary vascular resistance and the development of right heart failure [28]. According to CT findings, the volume of lung injury was significantly greater in non-survivor patients (80% [IQR: 64; 92] vs. 36% [IQR: 28; 48], $p < 0.001$).

Transthoracic echocardiography is important for the clinical assessment of patients with COVID-19, particularly those with moderate or severe disease, and it is necessary for monitoring patients with multiple areas of lung tissue consolidation in ARDS [29]. Analysis

of RV size, geometry, and function is an important component of cardiac assessment and contributes to clinical decision making in patients with cardiorespiratory failure [30].

Reservoir function of the right heart compensates increasing afterload by dilatation of both the RV and the RA. Regarding TTE findings, non-survivor patients had greater RV dilatation (basal RV diameter of 42 mm [IQR: 40; 48] vs. 39 mm [IQR: 37; 43] in survivor patients). Several authors have shown that RV dilatation in patients with COVID-19 was detected more frequently than systolic dysfunction, and adverse events were more common in these patients [31,32]. The increase of pulmonary vascular resistance is accompanied by RV dilatation and an increase of both the RV area and RV to LV area ratio. Although the RV/LV area values did not differ between groups ($p = 0.228$), this parameter demonstrated prognostic significance in the univariate analysis ($p < 0.01$) and in the multivariable model ($p < 0.03$).

Pagnesi et al. [32] assessed the prognostic value of pulmonary hypertension (PH) and RV dysfunction in hospitalized patients with COVID-19 infection ($n = 200$). According to the results of this study, PH was associated with severity of COVID-19 and with worse outcomes (all-cause mortality 33.3% vs. 6.3%, $p < 0.001$). PH may serve as a better predictor of cardiopulmonary changes in COVID-19 patients than RV dysfunction [32]. However, the subsequent increase in PH will be associated with decreased RV contractility due to the limited ability to adapt to the overload.

Our study revealed decreased systolic function in non-survivors, namely RV FAC (49.0% [IQR: 42.5; 53.1] in the non-survivors group vs. 53.4% [IQR: 46.4; 60.2] in the survivors group, $p = 0.007$), TAPSE (16 mm [IQR: 16; 19] vs. 20 mm [IQR: 19; 22], $p < 0.001$), tricuspid annular S wave (12 cm/s [IQR: 9; 13] vs.13 cm/s [IQR: 12; 15], $p = 0.016$), and RV FW LS (15.2% [IQR: 11.7; 18.3] vs. 22.3% [IQR: 17.7; 26.2], $p = 0.006$). Despite the significantly lower values of RV FAC in the non-survivors group, they were higher than the reference ones [10]. Bleakley et al. [33] showed that RV FAC can be used to identify patients with RV impairment and hypothesized that radial dysfunction and not longitudinal dysfunction is a dominant phenotype; however, this study was conducted in critically ill patients (VV ECMO proportion of 42.2%).

We found the association of TAPSE with mortality with OR = 0.572, 95% CI (0.429–0.764) ($p < 0.001$). In the meta-analysis by Martha et al. [34] the prognostic value of TAPSE in patients with COVID-19 has been shown, despite the fact that the decrease of this parameter in most studies exceeded the recommended value for the diagnosis of RV dysfunction [10]. This may be due to a number of limitations, such as dependence on scan angle, loading conditions, or overestimation of TAPSE in tricuspid regurgitation. In addition, this parameter reflects RV contractility mainly at the basal level. We noted a decrease of TAPSE usually in critical patients, but even among them, TAPSE values exceeded threshold values.

However, the use of conventional echocardiographic parameters has a limited value because of the complex shape of the RV [35,36]. It has been shown [36–38] that RV deformation analysis (STE) has a good predictive value as it has a higher sensitivity compared to the visual evaluation, and it can increase the predictive ability of conventional echocardiography even in presence of TR [39]. STE has been proposed for assessment of RV function due to an angle independency that leads to the increased precision in the RV dysfunction detection [10,35]. The independence of RV FW LS prognostic values from LV global systolic function in COVID-19 patients is one of the main advantages of this measure [36].

An important part of this study was the identification of adverse outcome predictors based on TTE data. The univariate analysis was performed by risk modelling (binary logistic regression) (Table 5). The identification of TTE risk factors for adverse outcome resulted in the construction of a multivariable prognostic model (1) ($p < 0.001$) for mortality risk prediction in patients with moderate to severe COVID-19. It included two risk factors for mortality: the sPAP and RV/LV area, and the preventive one (RV FW LS (2D STE)) (Figure 2).

Right ventricular abnormalities such as dilatation [40] and evidence of systolic dysfunction [36] have been reported as prognostic factors for adverse outcomes in patients with COVID-19. Dilation and systolic function alterations are most commonly associated with an increase in systolic pulmonary pressure [8]. We assume that the RV function changes may have a high predictive value in COVID-19, and a decrease in the ability of the RV to contract against increasing afterload leads to the development of disadaptation processes.

## 5. Limitations

The main limitation of this study is that it is a single-center study with a limited sample size. The significant heterogeneity of the data is due to large differences in populations, their ethnicity, and the lack of reference testing protocols. This work highlights the need for randomized controlled trials to confirm the results of the current study.

In addition, this study was limited to patients with moderate and severe COVID-19; patients with mild COVID-19 did not undergo TTE.

Finally, it is not possible to identify all the causes and pathogenetic mechanisms of cardiac pathology; therefore, the ability to predict the effect of COVID-19 on the occurrence, course, and prognosis of cardiovascular pathology is limited.

## 6. Conclusions

Patients with cardiovascular disease and COVID-19 have an increased risk of mortality and adverse outcomes. The ability to predict such outcomes may be a useful tool in disease management.

In our study, we showed that TTE protocol should be focused on the assessment of right heart size, RV contractile function, and pulmonary pressure as the most sensitive parameters of RV afterload and indirect markers of lung injury severity. The developed multivariable model incorporates parameters from standard and contemporary echocardiography and can be recommended for predicting the risk of adverse outcomes in patients with COVID-19.

**Author Contributions:** Conceptualization, E.Z.G. and I.V.S.; data curation, I.V.S., M.L.M., D.I.M., M.N.A. and I.V.V.; formal analysis, D.I.M.; investigation, I.V.S. and M.L.M.; methodology, E.Z.G., D.I.M., M.N.A. and M.M.R.; project administration, E.Z.G., M.M.R. and I.V.V.; visualization, I.V.S. and M.L.M.; writing—original draft, I.V.S.; writing—review and editing, E.Z.G., I.V.S., M.L.M., D.I.M., M.N.A., M.M.R. and I.V.V. All authors have read and agreed to the published version of the manuscript.

**Funding:** This research received no external funding.

**Institutional Review Board Statement:** The study was conducted in accordance with the Declaration of Helsinki and approved by the Ethics Committee of the A.N. Bakulev National Medical Research Center for Cardiovascular Surgery, Ministry of Health of the Russian Federation (protocol code 3 and date of approval: 25 November 2021).

**Informed Consent Statement:** Not applicable.

**Conflicts of Interest:** The authors declare no conflict of interest.

## Abbreviations

| | |
|---|---|
| TTE | transthoracic echocardiography |
| AUC | area under the curve |
| ROC | receiver operating characteristic |
| RV | right ventricle |
| LV | left ventricle |
| OR | odds ratio |
| sPAP | systolic pulmonary artery pressure |

| | |
|---|---|
| RV FW LS | right ventricle free wall longitudinal strain |
| 95% CI | 95% confidence interval |
| Se | sensitivity |
| Sp | specificity |
| ARDS | acute respiratory distress syndrome |
| CT | computed tomography |
| IQR | interquartile range |
| ICU | intensive care unit |
| COPD | chronic obstructive pulmonary disease |
| ASE | American Society of Echocardiography |
| EACVI | European Association of Cardiovascular Imaging |
| WHO | World Health Organization |
| TR | tricuspid regurgitation |
| LA | left atrium |
| Vol | volume |
| TDI | tissue Doppler imaging |
| FAC | fractional area change |
| EDA | end-diastolic area |
| ESA | end-systolic area |
| TAPSE | tricuspid annular plane systolic excursion |
| 2D STE | two-dimensional speckle-tracking echocardiography |
| PW | pulse-wave |
| RA | right atrium |
| MeanPAP | mean pulmonary artery pressure |
| BSA | body surface area |
| NEWS | patient severity rating scale |
| $SpO_2$ | blood oxygen saturation |
| EDI | end-diastolic volume index |
| ESI | end-systolic volume index |
| SI | stroke index |
| EF | ejection fraction |
| CI | cardiac index |
| ICC | intra-class correlation coefficient |
| AIRVO | humidifier with integrated air flow generator |
| VV ECMO | veno-venous extracorporeal membrane oxygenation |
| PA | pulmonary artery |
| PH | pulmonary hypertension |
| 95% CI | 95% confidence interval |

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
