# Peer review of "Transthoracic Echocardiography-Based Prediction Model of Adverse Event Risk in Patients with COVID-19"

_pathophysiology, doi:10.3390/pathophysiology29020014_

Round 1

Reviewer 1 Report

The manuscript is  overall interesting.

No abbreviations such as TTE should be used in the title.

In my opinion lines 81 -92 from Materials and methods should be moved to results. 

The authors must better explain why they  divided the patients in 2 groups: non-survivors and survivors.

Author Response

Good day! Thank you for your comments.

No abbreviations such as TTE should be used in the title. Corrected

In my opinion lines 81 -92 from Materials and methods should be moved to results. 

The "Results" section presents a comparison of the two groups, while lines 81-92 describe the total population under study. However, the description in section 2.1 is only for the main parameters, whereas the list of parameters to be compared is much longer in the Results section. With your permission, we would like to keep the information about the general population in the "Materials and Methods" section.

The authors must better explain why they divided the patients in 2 groups: non-survivors and survivors.

Thank you for your helpful comment! We have added this section to part 2.1. Study design and population profile (lines 94-95).

The study sample was divided into two comparable groups depending on the outcome of the disease: survivors (N=93) and non-survivors (N=17).

Reviewer 2 Report

Elena Zelikovna Golukova et al wrote a very interesting manuscript showing that values derived from transthoracic echocardiography, especially RV/LV area, sPAP and RV free wall longitudinal strain are significantly associated with mortality. I congratulate the authors to the manuscript. However some important information is missing:

  • During which timeframe was the study conducted? Which mutations of COVID-19 were present at this timeframe?
  • Has the institutional review board approved the study?
  • When did the patients die? I would recommend a Kaplan Meier curve. Please also add details of mortality into the text.

Minor comments:

  • Has long-term mortality been assessed?
  • Table 1: Which underlying rhythm did patients with PM have? I recommend to divide the cardiac rhythm into “PM without AF” and “PM with AF”.
  • Lines 115-116: It already has been stated previously that medical personnel was provided protective equipment.
  • I would rather move the baseline characteristics to the results section. This way, Table 1 is not needed independently, these data may be included in the remaining tables (with one additional column “all patients”).
  • The resulting logistic regression formula may be very inconvenient to use in daily clinical practice. How should clinicians in daily clinical practice proceed? Maybe the authors have a solution, for example a certain threshold of sPAP, RV/LV and RV strain above which mortality is significantly increased.

Author Response

  • Has the institutional review board approved the study?
  • Ethic Committee Name: Ethics Committee of the A.N. Bakulev National Medical Research Center for Cardiovascular Surgery, Ministry of Health of the Russian Federation
    Approval Code: â„–3
    Approval Date: November 25, 2021

  • When did the patients die? I would recommend a Kaplan Meier curve. Please also add details of mortality into the text.

An approximation of the survival of lost patients is possible using Kaplan-Meier curves when patients continue to meet the inclusion criteria even after dropping out of the study. That is, Kaplan-Meier curves are incorrectly applied because of impaired baseline preservation.

Unlike chronic diseases in which the disease continues to progress, infectious diseases break the basic principle of approximating the survival of lost patients - they no longer have COVID 19 and therefore no longer meet the inclusion criteria.

Minor comments:

  • Has long-term mortality been assessed?

Long-term mortality analyses have not yet been conducted; data collection for such analyses is continuing.

  • Table 1: Which underlying rhythm did patients with PM have? I recommend to divide the cardiac rhythm into “PM without AF” and “PM with AF”.

There were 3 patients in the study with pacemaker and they did not have atrial fibrillation. We made changes in Table 1.

  • Lines 115-116: It already has been stated previously that medical personnel was provided protective equipment.

To avoid repeating about personal protective equipment, we suggest replacing one sentence with another.

To reduce the risk of spreading infection, study time was limited [9].

  • I would rather move the baseline characteristics to the results section. This way, Table 1 is not needed independently, these data may be included in the remaining tables (with one additional column “all patients”).

We would like to avoid combining the tables as you suggest, because Table 1 presents data on the general patient population and only on basic parameters, while Table 2 compares the study groups on a much broader list of indicators. By combining the tables, we would have to give a lot of unnecessary information for readers; in our opinion, the combined table would be too cluttered.

  • The resulting logistic regression formula may be very inconvenient to use in daily clinical practice. How should clinicians in daily clinical practice proceed? Maybe the authors have a solution, for example a certain threshold of sPAP, RV/LV and RV strain above which mortality is significantly increased.

We presented the adjusted values for each individual predictor as OR, 95% CI, and p-value in Table 4. Calculation of individual threshold values for each quantitative predictor, determined, for example, by ROC analysis, we considered clinically inappropriate, because we consider a comprehensive approach to assessing the probability of a particular outcome to be much more reliable. So far, none of the ultrasound parameters has shown sufficient prognostic value in a single-factor analysis. This conclusion is confirmed by the lack of statistical significance of the RV/LV area parameter (p=0.228) when comparing groups of survivors and deceased, while its evaluation in the multivariable logistic model revealed a statistically significant effect.

The inconvenience of calculating the individual risk of mortality in patients using our model in routine medical practice can be easily eliminated with the help of simple commonly available software, for example, using formulas in MS Excel.

In the future, we are planning to create a special online program (calculator) to estimate mortality risk in patients with COVID-19 using our model.

Reviewer 3 Report

The authors made an interesting study with the aim to analyze patient survival and to develop a predictive model of survival in adults with 2019-nCoV infection based on transthoracic echocardiography parameters. They developed a prognostic model of mortality in patients with moderate and severe COVID-19 based on TTE data, which may have important clinical implications. I consider that this is generally a well-written and comprehensive article, with clear and legible tables and figures, and interesting findings that can make significant contributions to further large studies. I consider that the study is valuable and sound and can be published in its current form.

Author Response

Thank you so much for the work you have done and the positive review!